# Unveiling IL-33/ST2 Pathway Unbalance in Cardiac Remodeling Due to Obesity in Zucker Fatty Rats

**DOI:** 10.3390/ijms24031991

**Published:** 2023-01-19

**Authors:** Clementina Sitzia, Elena Vianello, Elena Dozio, Marta Kalousová, Tomáš Zima, Stefano Brizzola, Paola Roccabianca, Gabriella Tedeschi, John Lamont, Lorenza Tacchini, Massimiliano Marco Corsi-Romanelli

**Affiliations:** 1Department of Biomedical Science for Health, Medical Faculty, University of Milan, 20122 Milan, Italy; 2Institute of Medical Biochemistry and Laboratory Diagnostic, First Faculty of Medicine, Charles University and General University Hospital in Prague, 12108 Prague, Czech Republic; 3Department of Veterinary Medicine and Animal Science, University of Milan, 26900 Lodi, Italy; 4CRC “Innovation for Well-Being and Environment” (I-WE), University of Milan, 20122 Milan, Italy; 5R&D, RANDOX Laboratories Ltd., County Antrim BT29 4QY, UK; 6U.O.C. SMEL-1 of Clinical Pathology, IRCCS Policlinico San Donato, 20097 San Donato Milanese, Italy

**Keywords:** 3/10 max, IL-33/ST2, obesity, cardiac remodeling, cardiovascular disease, adipose tissue

## Abstract

Obesity is an epidemic condition linked to cardiovascular disease severity and mortality. Fat localization and type represent cardiovascular risk estimators. Importantly, visceral fat secretes adipokines known to promote low-grade inflammation that, in turn, modulate its secretome and cardiac metabolism. In this regard, IL-33 regulates the functions of various immune cells through ST2 binding and—following its role as an immune sensor to infection and stress—is involved in the pro-fibrotic remodeling of the myocardium. Here we further investigated the IL-33/ST2 effects on cardiac remodeling in obesity, focusing on molecular pathways linking adipose-derived IL-33 to the development of fibrosis or hypertrophy. We analyzed the Zucker Fatty rat model, and we developed in vitro models to mimic the adipose and myocardial relationship. We demonstrated a dysregulation of IL-33/ST2 signaling in both adipose and cardiac tissue, where they affected Epac proteins and myocardial gene expression, linked to pro-fibrotic signatures. In Zucker rats, pro-fibrotic effects were counteracted by ghrelin-induced IL-33 secretion, whose release influenced transcription factor expression and ST2 isoforms balance regulation. Finally, the effect of IL-33 signaling is dependent on several factors, such as cell types’ origin and the balancing of ST2 isoforms. Noteworthy, it is reasonable to state that considering IL-33 to have a unique protective role should be considered over-simplistic.

## 1. Introduction

Obesity is an increasing public health concern worldwide, as it is estimated that over 65% of the American adult population is overweight, and 42% are actually obese (NHANES, 2021). Indeed, obesity has been traditionally linked to cardiovascular disease development [1], but mechanisms that link fat accumulation and cardiovascular damage have not been fully understood.

Physiologically, visceral fat (VAT) as epicardial adipose tissue (EAT), secretes adipokines that exert protective effects on the myocardium, e.g., adiponectin [2] and orosomucoid [3] that mediate protection from apoptosis and antioxidant effects. For example, adiponectin improves nitric oxide bioavailability and endothelial function in the coronary vasculature [4]. In pathological conditions, VAT contributes to cardiac remodeling through the secretion of adipokines, which enhance low-grade inflammation and diabetic vascular complications. Indeed, EAT was shown to directly favor atherogenesis by activating monocytes [5] and through the deposition and oxidation of LDL [6]. In this sense, EAT thickness has been associated with ventricular myocardial mass increase (hypertrophy) and steatosis. Thus several studies have considered EAT as a quantifiable cardiovascular risk marker [7,8]. The cardiomyocyte’s damage could be mediated by the production of lipotoxic intermediates [9], such as retinol-binding protein-4 or activin A, which negatively affects cardiac metabolism [10]. Elegant work by Venteclef et al. showed that EAT secretome is enriched in pro-fibrotic adipokines [11]. In this respect, the ST2/IL-33 pathway has gained great importance in cardiovascular performance assessment. 

Interleukin 33 (IL-33), a member of the IL-1 family of cytokines, is an alarmin that induces a T helper-2 (Th2) skewed response [12]. In particular, IL-33, once released by necrotic cells or injured cells, acts in an autocrine/paracrine manner to activate the ST2 receptor. The ST2 ligand receptor (ST2L) is expressed on the membrane surface and activates the MAP kinase cascade and NF-kB to modulate an immune response. On the other hand, the soluble form of the ST2 receptor lacks the transmembrane and intracellular domains and functions as a decoy receptor, blocking IL-33 signaling [13,14]. In the cardiovascular system, IL-33 is highly expressed by endothelial, fibroblasts, or epithelial cells, while ST2L and sST2 mRNA are expressed at low levels in cardiomyocytes and cardiac fibroblasts, but they are highly expressed by the vascular endothelium. Finally, ST2L is also expressed in most immune cells. However, their transcription is inducible by pro-inflammatory signals—such as tumor necrosis factor (TNF)-α, interferon (IFN)-γ, and interleukin (IL)-1β—and specifically, IL-33 promotes sST2 transcription [14].

In cardiovascular disease, IL-33 secretion is considered to be protective since it reduces atherosclerotic plaque progression. On neonatal rat cardiomyocytes in vitro, it was demonstrated that IL-33 counteracted the effect of hypoxia by increasing the expression of anti-apoptotic proteins (XIAP, cIAP1, survivin, Bcl-xL). Furthermore, this effect was replicated in vivo, where exogenous IL-33 administration attenuated fibrosis, infarct size, and apoptosis after ischemia–reperfusion (I/R) and improved contractile function following coronary artery ligation in mice. These effects of IL-33 are ablated in ST2-/-mice, demonstrating that the beneficial influence of IL-33 is mediated by the interaction with its receptor ST2L [15]. Anti-inflammatory and protective roles of IL-33 involved adaptive (Treg expansion) and innate immunity (ILC2 regulation), however activation of mast cells or eosinophils due to exaggerated IL-33 signaling, could also promote fibrosis and inflammation, such in asthma and autoimmunity [16,17].

There are few works regarding the EAT-mediated expression of the IL-33/ST2 pathway. Of these, Gruzdeva et al. investigated the IL-33 system in relation to EAT thickness and visceral obesity in a cohort of post-AMI patients. They showed that EAT thickness increased with visceral obesity and correlated with fibrosis and maladaptive remodeling one-year post-AMI. The degree of cardiac fibrosis correlated negatively to the IL-33 levels and positively to the ST2 levels [18].

Indeed, a previous study of our group analyzed 55 patients presenting cardiovascular disease, including CAD and non-CAD patients, demonstrating an unbalance in IL-33/ST2 expression characterized by increased expression of ST2 in dysfunctional EAT tissue that correlated with left ventricle dysfunction [19]. The pathological EAT locally produced exchange proteins directly activated by cAMP-1 and 2 (EPAC1 and EPAC2) proteins, and the latter positively correlated with the expression of ST2 [19].

IL-33 signaling has also been linked to adipose tissue accumulation, as IL-33 is produced and released by adipocytes and endothelial cells in adipose tissue. Specifically, in obesity, adipose tissue is characterized by a pro-inflammatory phenotype. Inflammatory infiltrates mainly composed of classically activated M1 macrophages, cytotoxic T cells, and pro-inflammatory Th1-type cytokines (such as TNF-α and IFN-c). Those events are thought to be induced by hypoxia associated with the enlargement of adipocytes. Dyeing adipocytes recruit inflammatory cells and macrophages to clear cellular debris. In this context, IL-33 acts to modulate inflammation associated with obesity and to limit adiposity by increasing caloric expenditure. In endothelial cells of adipose tissue from severely obese subjects, elevated levels of both IL-33 and ST2 were observed [20]. Furthermore, ST2 knockout mice are more prone to develop obesity when fed a high-fat diet compared to wild-type mice [21].

Circulating levels of IL-33 and sST2 in obesity are controversial. Lower levels of serum IL-33 were observed in non-lean individuals in comparison to lean ones. IL-33 levels have also been negatively correlated to BMI in lean subjects, but this observation was not confirmed in obese subjects [22]. Other studies have observed increased serum levels of IL-33 in metabolically unhealthy overweight/obese subjects compared to healthy subjects. In addition, the increase in serum IL-33 was positively associated with several metabolic syndrome risk factors, such as diastolic blood pressure (DBP), and alanine transferase levels [23].

Obesity has been associated with increased plasma sST2 levels. Importantly several studies evidenced the prognostic utility of sST2 that predicts worse outcomes in acute myocardial infarction (MI) [24], heart failure (HF) [25], and type 2 diabetes mellitus (T2DM) [26]. Furthermore—in the long term—sST2 is an independent predictor of all-cause mortality. Moreover, the sST2 level could also predict left ventricular remodeling after myocardial infarction, being an important indicator of future cardiac function [27]. In this regard, the role of IL-33/ST2 pathway disbalance in obesity as a protective factor or negative mediator of cardiac remodeling has yet to be elucidated.

In this paper, we aimed to investigate cardiac remodeling in obesity, both in vivo and in vitro, and—in particular—to study the pathways linking ectopic fat accumulation and detrimental fibrotic remodeling of the heart. Here, we evaluated IL-33/ST2 expression and function related to cardiac fibrosis development in a rat animal model of obesity, the Zucker fatty rat. We demonstrated the deregulation of the IL-33/ST2 axis in both adipose and cardiac tissue and its effect on cardiomyoblasts gene expression both in vitro and in vivo, highlighting a new pathogenic role for fat accumulation in heart disease.

## 2. Results

### 2.1. Serum Levels of IL-33 and sST2 in Relation to Obesity in ZR

At sacrifice (25 weeks of age), Zucker fatty rats that harbor the homozygous mutation (Ob) developed obesity, and their weight was three times higher than those of controls (Lean) as previously reported [28] (Figure 1a). We measured IL-33 and the sST2 isoform levels in serum through ELISA, observing a decrease in IL-33 levels (Figure 1b) and a statistically significant increase in sST2 (Figure 1c). Intriguingly, we assessed a positive correlation between weight and IL-33 serum level in Lean rats. However, further increase in weight in Obese rats was associated with a decreased level of serum IL-33 (Figure 1d).

### 2.2. IL-33/ST2 Pathway Expression in Visceral Adipose and Cardiac Tissue

To investigate whether serum IL-33/sST2 was dependent on fat tissue abundance, we evaluated their amount in adipose tissue by ELISA: both IL-33 and total ST2 were increased in obese rats (Figure 2a,b respectively). However, ELISA analysis did not allow for discrimination between ligand and soluble ST2 isoforms. In cardiac biopsies, we observed a slight increase in IL-33 mRNA (Figure 2c) that became significant at the protein level, as shown in Figure 2f,h. However, we did not detect significant differences in ST2L or sST2 isoforms by means of qPCR (Figure 2d,e) or proteomic analysis. This finding is probably due to high variability in ST2 expression (Figure 2g,i,j) since ST2 isoform expression could depend on cell localization (as in atria or in ventricles) and on cell types.

### 2.3. Characterization of Fibrotic Signalling in Cardiac Tissue of Obese Rats

To investigate whether IL-33/ST2 deregulation determined fibrotic remodeling in obese hearts, we analyzed the expression of a panel of 84 genes involved in several fibrosis-related mechanisms, and we observed the activation of pro-fibrotic and tissue remodeling signaling. Accordingly, *Serpin1* is an inhibitor of tissue plasminogen activator and reduces the activity of matrix metalloproteinases, while *MMP8* and *MMP9* are peptidases involved in the degradation of the matrix. (Figure 3a).

These data were further confirmed by WB analysis (Figure 3b)—demonstrating significantly increased levels of TGFβ and collagen 3—and by Sircol analysis (Figure 3c), as we assessed increased collagen fibers deposition in cardiac biopsies from obese rats compared to Lean controls. Exchange protein activated by cAMP (Epac1) is an important mediator of cardiac remodeling, acting both as a pro-fibrotic [29] and an anti-fibrotic [30] agent, depending on cell type. Accordingly, we evaluated Epac1 expression in cardiac biopsies, and we observed its decrease both at mRNA (Figure 3d) and protein (Figure 3e,f) levels in Ob rats. Interestingly, the amount of Epac1 correlated negatively to IL-33 expression (Figure 3g).

In line with these results, we performed a morphologic analysis of cardiac tissues of obese rats to evaluate cardiomyocytes size and collagen deposition: although some areas of interstitial collagen deposition and adipocyte infiltrate (subepicardial) were detected in obese animals, these differences were not significant. Importantly, we did not have evidence of a significant remodeling of cardiac architecture (Figure 4a). Indeed, we performed immunohistochemical analysis of IL-33/ST2 expression in cardiac biopsies. According to the literature [31], we observed endothelial expression of IL-33/ST2—rarely in myofibers—however, we did not assess significant differences between Lean or Obese rats in protein localization or in the amount of protein expression (Figure 4b).

It is well accepted that altered IL-33 expression could determine an accumulation of T-reg cells in cardiac tissue, enhancing tissue repair and promoting immunosuppression [32]. Although the modulation of IL-33 expression in cardiac biopsies of ZF Obese rats, we did not observe a significant increase in Foxp3 protein expression (Figure 5), while lymphocyte accumulation was not assessed in cardiac tissue sections of Ob rats.

### 2.4. In Vitro Characterisation of Cardiomyocytes following VAT Stimulation

To better elucidate the interplay among adipose tissue-derived signals and IL-33/ST2 expression in cardiac cells, we performed transwell experiments by plating H9C2 cells with VAT tissue derived from Lean or Obese animals, and we measured adipose tissue-derived IL-33 in coculture experiment. ELISA performed on cell medium at 6 h demonstrated a higher amount of IL-33 expression by adipose tissue from Zucker rats than those from lean rats (*p* = 0.0019; Appendix A). Following these experimental evidence, we suggested that adipose tissue could release IL-33 and provoke the further release of IL-33 itself by cardiomyocytes. To further support our hypothesis, we assessed the expression of the IL-33/ST2 pathway by qPCR, and we found that adipose tissue stimulation determined the increase in IL-33 expression in H9C2 cells compared to control cells (Figure 6a). More interestingly, we observed a significant increase in IL-33 expression in cells cultured with adipose tissue from Ob rats related to those cultured with adipose tissue from Lean rats (Figure 6a).

Conversely, stimulation of adipose tissue with obese-derived VAT determined a slight reduction in the expression of ST2L while soluble sST2 did not vary, suggesting deregulation in IL-33/ST2L pathway due to obesity (Figure 6b,c). Finally, Epac1 expression was reduced in cells cultured with obese-derived VAT—in line with data obtained in cardiac biopsies—and negatively correlated to ST2L expression (*p* < 0.001) (Figure 6d,e).

We further analyzed whether fat stimulation could promote fibrotic or hypertrophic signaling in H9C2 cells by analyzing Yin Yang 1 (YY1) and Myocyte-specific enhancer factor 2A (Mef2A), according to their role in ST2 isoforms’ regulation [33,34,35] and hypertrophic response [34], respectively.

We observed a significant reduction in *YY1* (Figure 6f) due to VAT stimulation independently from the provenance of VAT tissue and—similarly—a reduction in *Mef2A* YY1 (*p* < 0.01) (Figure 6g). The reduction in *YY1* was directly related to the reduction in *Epac1* signaling (Figure 6h). These data suggest that adipose tissue promoted a reduction in hypertrophic/fibrotic signaling mediated by IL-33-mediated repression of Epac1 transcription. Indeed, ST2L expression was also negatively related to *YY1* expression suggesting that the reduction in ST2L is linked to a switch towards a fibrotic response (Figure 6i).

### 2.5. In Vitro Ghrelin Stimulation of Cardiomyocytes Modulated IL-33/ST2 Expression

It is well accepted that the lack of leptin signaling in Zucker fatty rats determines an increase in the level of serum ghrelin (GHR), a hormone principally secreted by gastric mucosa but also by cardiomyocytes, that normally bear GHR receptor [35]. Thus, to reproduce the ZF condition in vitro, we assessed the effect of GHR stimulation on cardiomyocytes and how GHR signaling affected IL-33 release. We treated H9C2 cells with GHR for 72 h, and we observed by means of qPCR that, starting from 48 h of treatment, GHR stimulated IL-33 expression in H9C2 cells and up-regulated *Epac2* that in adipose tissue controls adipogenesis [36] (Figure 7a–c). We further observed the up-regulation of *Mef2A* (Figure 7d) and *HDAC4* (Figure 7e) that, together with *YY1,* can stimulate sST2 transcription, but we did not observe changes in *YY1* expression (Figure 7f). Indeed, we assessed the increased expression of *GHR receptor (GHSR)* (Figure 7g), and—interestingly—we showed an up-regulation of *sST2* (Figure 7i) but not of *ST2L* (Figure 7h).

### 2.6. Characterization of Ghrelin-IL-33/ST2 Axis in Zucker Rat Cardiac Tissue

As we observed a GHR-dependent increase in IL-33 expression in cardiomyocytes in vitro—that was linked to up-regulation of *sST2* and *Mef2A*—we assessed the mRNA levels of all these targets in cardiac biopsies of ZR to verify that this pathway is active in vivo. Unexpectedly, we did not observe relevant differences in the expression of *YY1* or *Mef2A* in cardiac biopsies of Ob versus Lean rats (Figure 8a,b). However, the expression of these two factors was slightly positively correlated (Figure 8c). In heart homogenates, the increase in GHSR expression confirmed the activation of GHR signaling in Ob rats (Figure 8d).

Finally, we evaluated that leptin stimulation of H9C2 cells after 4 h or 48 h reduced IL-33 expression in cardiomyocytes with an opposite trend than those observed after GHR stimulation. These data demonstrated that adipose tissue and its secretome are directly involved in IL-33 signaling modulation in cardiomyocytes and that they could affect gene expression in cardiomyocytes (Appendix A).

## 3. Discussion

Obesity is a multifactorial condition characterized by excessive fat accumulation. Although there is a well-known association between obesity and cardiovascular pathology, the mechanism by which fat accumulation provokes heart remodeling has not been definitely understood. Ectopic fat accumulates in the heart while circulating adypokines are able to reach the myocardium, influencing its metabolism and gene expression. In this sense, IL-33 signaling is traditionally considered to exert a protective role in cardiovascular remodeling through the interaction with its receptor ST2L [13,15,37]. On the contrary, the soluble form of the ST2 receptor blocks its protective functions. However, some controversies exist, and the role of IL-33 in obesity is still a matter of debate [23,38]. Here we analyzed the mechanisms by which obesity alters IL-33/ST2 regulation and how this unbalance could affect myocardium gene expression.

Firstly, we showed that obesity determined a dysregulation of the IL-33/ST2 pathway characterized by an opposite relation between weights and circulating IL-33 levels in Lean versus Obese animals. Indeed IL-33 seems to increase with body mass in lean animals, but it tends to decrease when fat accumulates in obese rats. However, the measurement of IL-33 in serum suffers from some limitations, such as short-half life; rapid clearance from the circulation by proteases; oxidization by reactive species. Notably, IL-33 availability depends on its binding to sST2, whose concentration is increased in obese rats: it is possible that the real amount of circulating IL-33 in obese rats was underestimated. On the other hand, the assay of sST2 is considered reliable, and it is now included in clinical workup for cardiovascular risk assessment. Thus, sST2 increases in ZF deponed for ventricular dysfunction due to obesity. Accordingly, IL-33 increased also in cardiac and VA tissue of obese rats.

Unfortunately, we failed to demonstrate a significant difference in ST2 cardiac isoforms’ expression, possibly as a consequence of the differential ST2 expression depending on cell type (cardiomyocytes, cardiac fibroblast, endothelium, or immune cells) and localization.

As we observed a dysregulation in IL-33/ST2 signaling, we determined in cardiac biopsies an increased expression of genes that regulates matrix degradation, indicating fibrotic remodeling in obese rats [39]. Although these findings were further confirmed by increased TGF-β and collagen3 expression, histological analysis showed comparable levels of interstitial collagen deposition without significant modification of cardiac architecture. Similarly, immunohistochemical analysis of cardiac tissue did not show enriched endothelial expression of IL-33 or Treg expansion, suggesting that the effect of increased IL-33 predominantly involved cardiomyocyte signaling. Anyway, we did not further evaluate the Th2 response.

Among the mediators of cardiac remodeling, we focused on Epac proteins (Epac 1 and 2) [40,41], as we previously determined a positive correlation between Epac2 and IL-33 signaling in EAT tissue [19]. Epac1 hypertrophic signaling could be mediated by CaMKII regulation of nuclear export of HDAC4/5 [42] that allows the activity of the pro-hypertrophic transcription factor MEF2A [43]. Following these premises, in obese Zucker rats, we found reduced expression of Epac1 negatively correlated to IL-33 expression, suggesting a possible modulation of TGF-β/MEF2A regulation of transcription.

To determine whether adipokines from adipose tissue could directly influence cardiomyocyte metabolism, we performed a transwell coculture of H9C2 cells with VAT derived from Lean or Obese Zucker rats. Firstly, we observed that conditioned medium from Obese rats contained a higher level of IL-33 and, accordingly, IL-33 expression was up-regulated in conditioned H9C2 cells. On the other hand, ST2L expression was greatly increased following stimulation from Lean animals. In this regard, we hypothesized that circulating IL-33 in vivo in Obese rats is predominantly derived by pathological fat accumulation. Finally, in Zucker rats, Epac1 expression was reduced by fat stimulation and negatively correlated to ST2L expression. According to these results, we demonstrated that IL-33/ST2 signaling is deregulated in obesity and, importantly, altered ST2L/sST2 balancing drove the pro-fibrotic switch in gene expression. Furthermore, we hypothesized that fat stimulation could influence the expression of YY1 and MEF2A, modulating hypertrophic or fibrotic response. The complex mechanism that drives the interplay between YY1/MEF2A and IL-33/ST2 is not understood, and further knock-out studies are needed to clarify this point. Interestingly, it is reported that YY1 can increase the transcription of sST2 through a consensus site in its promoter region, directly involved in the fibrotic remodeling of perinfarctual zones [44]. YY1 has also been involved in obesity by promoting triglyceride accumulation—and it is up-regulated in the hepatic tissue of both obese animals and NAFLD patients [45].

Zucker rats present a lack of leptin signaling—an hormone involved in satiety and food intake but also in the regulation of energy homeostasis and skeletal growth [46]—that determines an increase in serum GHR level. Since GHR is secreted by cardiomyocytes which normally present GHSR, we decided to assess whether GHR signaling could affect IL-33 release in vitro. We observed that GHR stimulated IL-33 expression in H9C2 cells and upregulated MEF2A, whose regulation is also Epac-dependent. Interestingly we showed an upregulation of sST2 that can be mediated by HDAC4/YY1 stimulation. On the other hand, preliminary data on leptin stimulation of H9C2 cells showed an opposite effect than GHR treatment suggesting that the balance of these two hormones affects IL-33 signaling.

Next, we combined the effects of fat stimulation and GHR signaling on cardiomyocytes in vivo. The up-regulation of GHSR in ZF-obese rats demonstrated the activation of this signaling, but we did not observe any difference between MEF2A and YY1 expression. These findings are probably the result of the GHR compensative pathway that counteracts the dysregulation of IL-33/Epac signaling.

## 4. Material and Methods

### 4.1. Animal Models Included in This Study

Obesity in Zucker rats arose from a spontaneous mutation (a missense A to C mutation) in the leptin receptor gene on chromosome 5 (Leprfa) that determines the production of a non-functional receptor. The mutation leads to a state of hyperphagia due to an impaired satiety reflex that leads to obesity [47].

Procedures involving living animals were conformed to Italian law (D.L.vo 116/92), to the European Union guidelines, and approved by local ethics committees. Ten obese non-diabetic male Zucker rats (OB) (fa/fa-) and 10 lean littermates (L) (Fa/+) were provided by Charles River. All animals were housed in a controlled ambient environment (12 h light/dark cycle) at a temperature between 21 and 23 °C. Cage population was limited to a maximum of four animals each to ensure the health and welfare of animals. The rats had free access to clean water and were fed the standard diet. At 25 weeks of age, rats were deeply anesthetized with Zoletyl (20 mg/kg), then sacrificed by cervical dislocation. One animal of the Lean group died prior to the study endpoint died thus, the organs were not collected. Heart, adipose tissue, and serum were collected. For each group, 5 hearts were formalin-fixed and paraffin-embedded, 5 hearts, and 5 adipose tissue biopsies. Were frozen in Allprotect Tissue Reagent (QIAGEN, Hilde, Germany) at −20 °C. Mutated Zucker rats were called the Obese group, while wild-type healthy rat was called the Lean. We performed and analyzed multiple biopsies of frozen hearts to increase the power of statistical analysis.

### 4.2. Total RNA Extraction and Reverse Transcription

Disruption and homogenization of cardiac and VAT samples (*n* = 5 for each experimental group) were performed with the TissueLyser II* equipment (QIAGEN) through high-speed shaking in plastic tubes with 5 mm stainless steel beads. Then, total RNA was isolated using the RNeasy Lipid Tissue Mini Kit (QIAGEN), according to the manufacturer’s instructions. RNA concentration was quantified with NanoDrop (Thermo Fisher Scientific, Waltham, MA, USA). RNA samples (1 μg) were first treated with a genomic DNA elimination step (5 min/42 °C and kept on ice for at least 1 min) and then reversely transcribed using the RT2 First Strand Kit (15 min/42 °C and 5 min/95 °C) (QIAGEN). Samples were stored at −20 °C. Coculture cells were harvested from 48 wells centrifuged, and cells pellet were frozen at −80 for further use. RNA was extracted from harvest cells with RNeasy Microkit (Ref 74004 QIAGEN). cDNA was generated using the Reverse Transcriptase (Kit RT2 First Strand Kit Cat. No./ID: 330404) followed by the SYBR-Green reaction to quantify the expression of the genes in Table 1 (, RT2 Sybr green qPCR Mastermix fast fromQiagen, Hilden, Germany). All the cDNA samples were tested in triplicate, and the threshold cycles (Ct) of target genes were normalized against a housekeeping gene, the glyceraldehyde 3-phosphate dehydrogenase (GAPDH). Relative transcript levels were calculated from the Ct values as X = 2^–ΔΔct^ where X is the fold difference in the amount of target gene versus GAPDH and ΔΔCt = ΔCt_target_ − ΔCt_GAPDH_. The efficiency of primers used was calculated between 95.2% and 98.3%. A list of Custom LNA Oligonucleotide is included in Table 2. 

### 4.3. RT2 Profiler PCR Arrays

RT2 Profiler PCR Arrays allowed the detection of 84 key gene transcripts related to rat fibrosis (PARN-120Z, from QIAGEN, Hilden, Germany) using qPCR. In particular, these genes were related to the regulation of extracellular matrix (ECM), cell adhesion molecules, and in fibrogenesis, including both pro- and anti-fibrotic regulators, growth factors, inflammation-related molecules, signal transduction molecules, and regulators of epithelial-to-mesenchymal cell transition.

Each cDNA sample was diluted with nuclease-free water and mixed with the RT2 SYBR green Mastermix (QIAGEN, Hilden, Germany). Twenty-five μL of the same experimental mixture was automatically added to each well of the array (one array for each cDNA) using the QIAgility^®^ equipment (QIAGEN, Hilden, Germany). qPCR was performed by the RotorGene-Q (QIAGEN, Hilden, Germany) and consisted of an initial activation of the Hot-start DNA Taq polymerase at 95 °C/10 min, which was followed by 40 cycles of 95 °C/15 s and 60 °C/30 s. Then, dissociation curves were performed to verify the specificity of the amplicons using the default melting curve program of the instrument. Data were analyzed using the RT2 Profiler PCR Array Data Analysis Web Portal (QIAGEN, Hilden, Germany A list of all the transcripts included in the fatty acid metabolism array and in the fibrosis array can be found in the manufacturer manual (QIAGEN, Hilden, Germany).

### 4.4. Western Blot Analysis

Total proteins from cardiac biopsies isolated from lean and obese rats were extracted in RIPA (Cell Signaling, Danvers, MA, USA) lysis buffer 1x + 1% protease/phosphatase inhibitor (Cell Signaling) + dH₂0, then disrupted and homogenized with TissueLyser II* (QIAGEN, Hilden, Germany). Samples’ homogenization was carried out in high-speed (20–30 Hz) shaking steps for 2 min for 2 consecutive cycles (in 1.2 mL microtubes containing 5 mm stainless steel beads), then put on ice for 30′ and centrifuged (at 4 °C −13,200 rpm for 15′) to collect the supernatant as described in QIAGEN Tissue Lyser Handbook. Cells were incubated with 250 μL of Trypsin/EDTA for 5 min at 37 °C. As soon as lysis occurs, proteolysis, dephosphorylation, and denaturation begin. Trypsin was inactivated, and lysis was stopped treating the cells with 250 μL of cell culture medium. The total volume was transferred to a microfuge tube, and the lysate was clarified by spinning for 8 min at 1200 rpm at room temperature. The cell medium was discarded, and the cellular pellet was immediately frozen at −80 °C. Samples were resolved on polyacrylamide gels (Mini-Protean TGXStain–Free Gels ranging from 4% to 20%) (BIO-RAD, Hercules, CA, USA) and transferred (transfer system BIO-RAD Trans-Blot Turbo) to nitrocellulose membranes (BIO-RAD Trans-Blot Turbo 0.2 µm nitrocellulose all from Bio-Rad Laboratories. Membranes were incubated with primary antibodies ON at 4 °C, followed by washing, detection with horseradish peroxidase (HRP)-conjugated secondary antibodies (DakoCytomation, Santa Clara, CA, USA), and developed by enhanced chemiluminescence (ECL BIO-RAD ClarityTM Western substrate). Filters were incubated overnight. A list of all tested antibodies is included below. Bands were visualized using BIO-RAD Chemidoc™ Touch Image System. Densitometric analysis was performed using ImageJ software Version 1.53t.

### 4.5. Cell Culture and Transfection

The rat H9C2 cell line was utilized in this study. All the cells were obtained from H9C2 Merck Cell Line from Rat BDIX heart myoblast-88092904. The cells were grown in DMEM high glucose (Euroclone) medium supplemented by 10% fetal bovine serum (FBS) and incubated at 37 °C, with a 5% CO_2_ atmosphere. For transwell assay, cells were plated in a 24-well plate (30,000 cells/wells) with transwell insert (Costar code 5026649) in 0.5 mL of medium for each experimental time-point. 30 mg of fresh adipose tissue was plated on the inner chamber, on the top of the filter membrane, while cells were plated on the bottom of the lower chamber in a 24-well plate (30,000 cells/wells). The peculiar pore size of the transwell membrane employed in our experiment did not allow direct cell-cell interaction between adipose tissue and H9C2, nor did the passage of adipose cells. However, the media of H9C2 were continuously conditioned by the secretome of VAT cells. Cells and medium were collected after 3 h, 6 h, 24, and 48 h, and RNA extraction was performed on lysed cells through a Qiagen kit. H9C2 cells were seeded for Ghrelin treatment (Ghrelin rat lyophilized powder G8902) (Sigma-Aldrich, St. Louis, MO, USA) in 24 well-plate (FALCON 353504) seeding density 30,000 cells in 0.5 mL of medium/well, treated at T0 and T48h and harvested every 24 h till 72 h. H9C2 cells were also seeded for Leptin treatment (Leptin rat lyophilized powder L5037-Sigma-Aldrich) in 24 well-plate (FALCON 353504) seeding density 30.000 cells in 0.5 mL of medium/well, treated at T0 and harvested at T4h, T24h till 48 h.

Following harvesting, cells were immediately lysed in Buffer RLT to prevent unwanted changes in the gene expression profile, as suggested by Qiagen RNeasy Micro Kit protocol.

### 4.6. ELISA Assay

ELISA assays were performed to analyze protein expression in serum and VAT tissue of ZF rats. Rat IL-33 (Fine Test, Wuhan, Hubei, China) for IL-33 analysis and Rat IL1RL1 (IL-1 Receptor Like1) (Fine Test, Wuhan, Hubei, China) for ST2 was performed. Serum and VAT tissue homogenates were prepared following kit instructions. Serum was diluted 1:2 before assay testing.

### 4.7. Sircol Assay

The collagen amount was quantified in homogenates of cardiac biopsies by Sircol Collagen Assay (Sircol™, Soluble collagen assay Biocolor Ltd., Carrickfergus, Co Antrim UK). It measures mammalian collagen (I to V). The procedure was performed following kit instructions. 50 μL of cardiac homogenates were analyzed in this assay.

### 4.8. Histological Analysis

For each group, 5 hearths were formalin-fixed, paraffin-embedded, and cut on a microtome. Hematoxylin and Eosin and Trichrome Masson stains (Trichrome Stain Masson, Sigma-Aldrich St. Louis, MO, USA) were performed on 0.5 mm serial sections of cardiac biopsies from ZF rats. Immunohistochemical Staining of IL-33 and ST2 was performed through the automated procedure Ventana Medical Systems (Roche Basel, Switzerland). Antibodies were used as follows: IL-33 Rabbit monoclonal 1:200 (Proteintech, Thermo-Fisher Group Waltham, MA, USA) and ST2 Rabbit (Enzo Life Sciences, Euroclone, Pero, Milan, Italy) polyclonal antibody 1:500. Images analysis was performed by the Anatomic Pathology Service of the Department of Veterinary Medicine and Animal Science of the University of Milan.

### 4.9. List of Included Antibodies

Collagen 3 alpha1 antibody, Cohesion biosciences (London, UK), mouse monoclonal 1:1000 140 KDa.IL-33 Enzo, rabbit polyclonal antibody, 1:1000 25 kDa.Epac 1 (5D3), Cell signaling, Mouse mAb 1:1000 100 kDa.Epac 2 (5B1), Cell signaling Mouse, mAb 1:1000 115 kDa.FOXP3 (F-9) Santa Cruz (Dallas, Texas) Mouse monoclonal 1:200 48 kDa.ST2, Proteintech (Rosemont, Illinois, USA), polyclonal antibody Rabbit 63/37/30 kDa.ST2, Enzo, polyclonal antibody, 1:500.Tgf beta1, Cohesion biosciences, rabbit polyclonal, 1:500 43 kDa.Vinculin, Cell Signaling, Rabbit mAb 1:1000 124 kDa.

### 4.10. Statistics

Data were analyzed by GraphPad PrismTM and expressed as mean ± SD or mean ± SEM. To compare multiple groups’ means, one-way ANOVA followed by Tukey’s multiple comparison tests was used to determine significance (* *p* < 0.05, ** *p* < 0.01, *** *p* < 0.001; **** *p* < 0.0001). In order to compare the two groups, Student’s *t*-test was applied assuming equal variances: the difference was considered significant at * *p* < 0.05. To correlate the protein’s expression, linear regression and multivariate regression analyses were performed.

## 5. Conclusions

In conclusion, we showed that obese adipose tissue promoted IL-33 production both in vivo and in vitro. In our model, we described that GHR up-regulation determines IL-33 over-expression. However, in the literature, conflicting results have been reported regarding the amount of IL-33 in obesity as it depends on the animal model and the type of tissue analyzed or, alternatively, on cellular and temporal expression (acute or chronic) [13]. In fact, exogenous short administration of IL-33 seems to be protective in the obesogenic condition in mice [48], while IL-33 up-regulation in healthy mice could determine pericarditis [49].

In vitro experiments demonstrated that fat accumulation determined ST2L reduction, suggesting how cardioprotection is lost despite the IL-33 increase. Furthermore, GHR stimulation enhanced sST2 production rather than ST2L. Indeed, both GHSR and sST2 are involved in hypertrophic response related to the stretch of cardiomyocytes, and they are both considered a marker of heart failure since they increase with the severity of ventricular dysfunction in human disease [50]. On the other hand, GHR also protects the heart by reducing apoptosis and promoting the survival of cardiomyocytes through the up-regulation of MEF2A. Indeed, although Obese Zucker rats presented a molecular signature of activated fibrotic response to fat accumulation, this condition did not alter cardiac architecture due to the compensative response of IL-33 and GHR. Thus, the effect of the IL-33 signal in our model seems to be protective and increased by GHR stimulation, but its activation also promoted sST2 expression, counteracting its protective functions.

The opposite effects of IL-33 resembled those observed in liver fibrosis, where it is tissue-protective when acute, whilst it functions as a pro-fibrotic hepatic factor in cases of chronic injury [51].

Interestingly IL-33 signaling could be referred to as the “obesity paradox,” as being overweight or obese is associated with a favorable prognosis while having increased cardiovascular risk. Similarly, an overweight condition would promote IL-33 signaling and ameliorate low-grade inflammation while increasing cardiovascular risk through increased sST2. Our data suggest that considering IL-33 as solely protective is probably over-simplistic. Furthermore, to solve this dichotomous role of IL-33, it is necessary both to evaluate the cellular source of the protein and to distinguish its effects on different pathways by conducting specific genetic deletion studies.

In this exploratory paper, we proposed a role for Epac proteins in heart remodeling and their link to IL-33/ST2 pathway, but further experiments combining the employ of analog or inhibitor of Epac proteins are needed to properly decipher their functions. In conclusion, we improved the knowledge on IL-33/ST2, suggesting the involvement of these proteins in the obesity axis, and we highlighted for the first time a modulation driven by ghrelin—paving the way for future mechanistic studies.

## Figures and Tables

**Figure 1 ijms-24-01991-f001:**
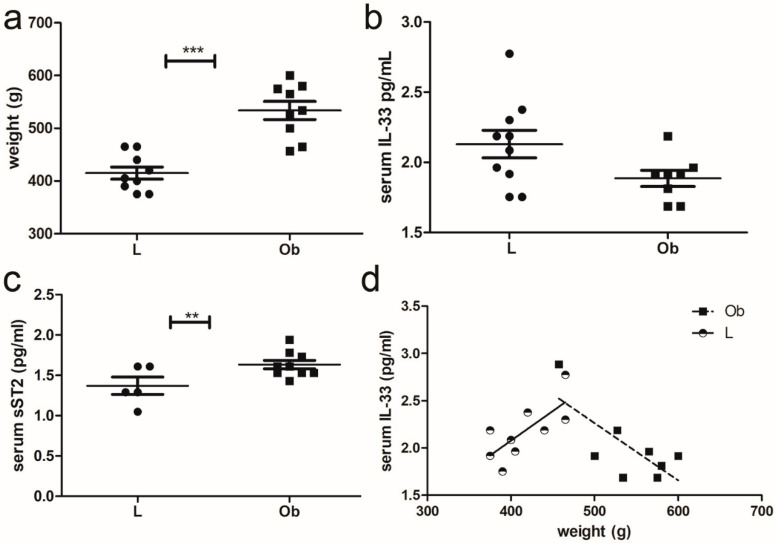
Characterization of IL-33 signaling in ZF rats: (**a**) Graphs showing the weight of rats at sacrifice (*** *p* < 0.001), (**b**) ELISA assay of serum IL-33. (**c**) ELISA assay of ST2 serum levels and (**d**) graph showing an opposite correlation between IL-33 and weight in Lean (** *p* < 0.01) versus Obese rats. Student’s *t*-test: ** *p* < 0.01; *** *p* < 0.001. Full square stands for Obese rats, while full circle stands for Lean rats.

**Figure 2 ijms-24-01991-f002:**
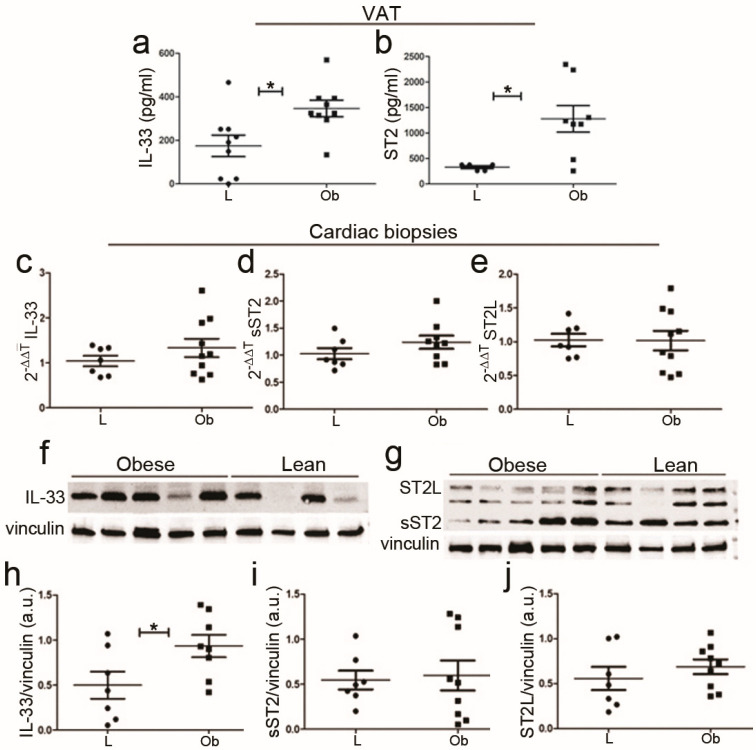
IL-33/ST2 expression in VAT tissue and cardiac biopsies of ZF rats. Graphs showed up-regulation of IL-33 (**a**,**b**) ST2 expression in visceral adipose tissue (VAT). Following qPCR experiments (**c**–**e**), graphs showed *IL-33*, *sST2*, and *ST2L* expression in cardiac biopsies. (**f**,**g**) Representative blots showed increased IL-33 protein and no difference in ST2 isoform levels (densitometric analysis (**h**–**j**)). Student’s *t*-test: * *p* < 0.05. Full square stands for Obese rats, while full circle stands for Lean rats.

**Figure 3 ijms-24-01991-f003:**
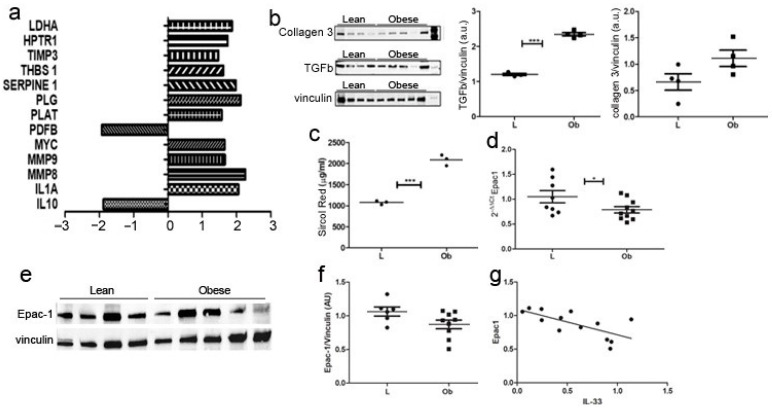
Fibrotic cardiac remodeling evaluation in obese Zucker rat. (**a**) The graph showed fibrosis-related transcriptome in ZF rats. Genes that showed a fold change >2 or <−2 in the Ob vs. L group are represented and grouped according to their biological function (*p* < 0.01). (**b**) Western blot showed upregulation of collagen 3 and TGF-β in Obese rats. (**c**) Graph showing Sircol red quantification demonstrated significant over-expression in Obese rats related to control ones (*p* = 0.0002). Graphs showed *Epac-1* over-expression in (**d**) qPCR and (**e**) proteomic experiments, as shown by representative blot and (**f**) densitometric analysis. (**g**) A negative correlation was evidenced between Epac1 and IL-33 expression Student’s *t*-test: * *p* < 0.05; *** *p* < 0.001. Full square stands for Obese rats, while full circle stands for Lean rats.

**Figure 4 ijms-24-01991-f004:**
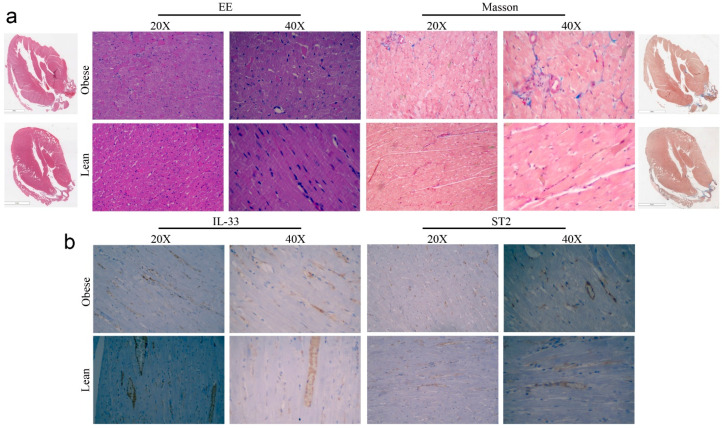
Pictures of (**a**) Haematoxylin and Eosin and Masson’s staining and (**b**) IHC analysis of IL-33 and ST2 expression in cardiac biopsies of ZR.

**Figure 5 ijms-24-01991-f005:**
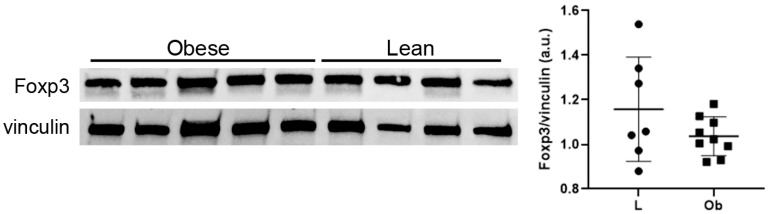
Western blot showed no difference in Foxp3 expression levels. Student’s *t*-test. Full square stands for Obese rats, while full circle stands for Lean rats.

**Figure 6 ijms-24-01991-f006:**
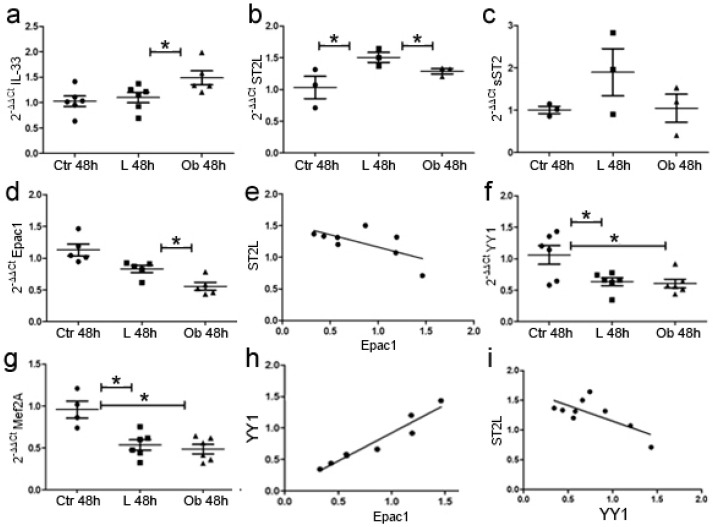
Conditioned medium with VAT tissue promotes cardiomyocytes’ gene expression alteration and hypertrophic/fibrotic signaling. qPCR experiments showed that fat stimulation from Obese rats promoted (**a**) *IL-33* expression while it reduced (**b**) *ST2L*. (**c**) *sST2* was not modified. Fat stimulation reduced (**d**) *Epac1* expression. (**e**) The graph showed a negative—but not significant—correlation between ST2L and Epac1 expression. Fat stimulation reduced (**f**) *YY1* expression in both Lean and Ob and (**g**) *Mef2A* expression in both Lean and Ob versus CTR. (**h**) The graph showed a positive correlation between YY1 and Epac1 while (**i**) a negative correlation emerged between ST2L and. Student’s *t*-test: * *p* < 0.05. Full square stands for Obese rats, while full circle stands for Lean rats.

**Figure 7 ijms-24-01991-f007:**
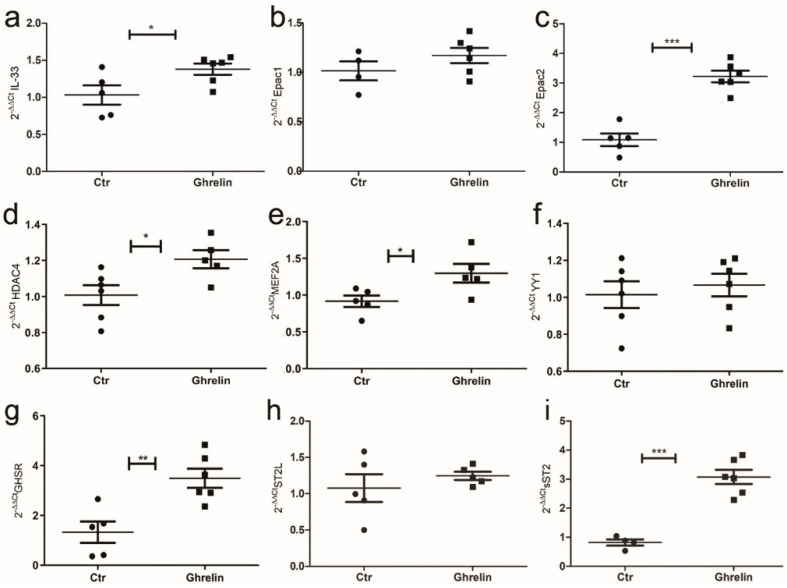
Ghrelin in vitro treatment of H9C2 cells modulated IL-33/ST2 signaling. Graphs showed qPCR of H9C2 cells treated with ghrelin or vector at 72 h. Ghrelin treatment enhanced (**a**) *IL-33* and (**c**) *Epac2* expression, while (**b**) *Epac1* was unchanged. (**d**) Ghrelin treatment also induced *HDAC4* and (**e**) *MEF2A,* while it did not alter *YY1* expression (**f**). Ghrelin treatment enhanced (**g**) *GHSR* expression, while (**h**) *ST2L* was unaffected. (**i**) The *sST2* mRNA was greatly increased by ghrelin treatment. Student’s *t*-test: * *p* < 0.05; ** *p* < 0.01, *** *p* < 0.001. Full square stands for Obese rats, while full circle stands for Lean rats.

**Figure 8 ijms-24-01991-f008:**
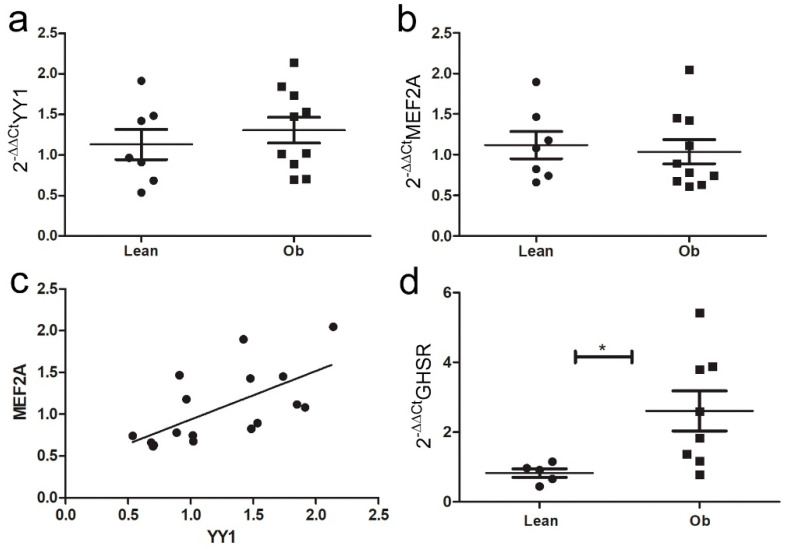
Ghrelin-induced pathway analysis in ZR cardiac biopsies. Graphs showed qPCR of (**a**) *YY1* and (**b**) *MEF2A* expression in cardiac biopsies of ZR: (**c**) a positive correlation was observed between MEF2A and YY1 in cardiac biopsies of ZR. (**d**) *GHSR* mRNA was greatly increased in vivo in obese ZR (* *p* < 0.05). Student’s *t*-test: * *p* < 0.05. Full square stands for Obese rats, while full circle stands for Lean rats.

**Table 1 ijms-24-01991-t001:** RT2qPCR Primer Assay QIAGEN.

Prime Name	Primer Code	Amplicon Length
Rat Yy1	PPR53391A	99 bp
Rat Socs3	PPR06602A	145 bp
Rat Rapgef3 (Epac1)	PPR49530A	131 bp
Rat Rapgef4 (Epac2)	PPR49522A	119 bp
Rat Pparα	PPR44459A	110 bp
Rat Mef2A	PPR62504B	100 bp
Rat IL33	PPR564110A	97 bp
Rar Hdac4	PPR47615A	89 bp
Rat Ghsr	PPR51984A	121 bp
Rat Ghrl	PPR49492A	107 bp
Rat Gapdh	PPR06557B	200 bp

**Table 2 ijms-24-01991-t002:** Custom LNA oligonucleotide (QUIAGEN).

Prime Name	Primer Code	Amplicon Length
Forward Rat St2L	AGTTGTGCATTTACGGGAGAG	68 bp
Reverse Rat St2L	GGATACTGCTTTCCACCACAG	
Forward Rat St2s	GGTGTGACCGACAAGGACT	119 bp
Reverse Rat St2s	TTGTGAGAGACACTCCTTAC	

## Data Availability

The data presented in this study are available on request from the corresponding author.

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
