# Peer review of "Unveiling IL-33/ST2 Pathway Unbalance in Cardiac Remodeling Due to Obesity in Zucker Fatty Rats"

_ijms, 2023, doi:10.3390/ijms24031991_

Round 1

Reviewer 1 Report

Sitzia et al. study cardiac remodeling in obesity. Specifically, using both in vivo and in vitro models, they investigated the role of IL-33/ST2 axis in adipose tissue and cardiac tissue and the possible involvement in detrimental fibrotic remodeling of the heart. In addition, the authors proposed a modulation driven by ghrelin.

The manuscript is well written and easy to follow. This is an important study in an area of intense interest; however, elucidation of some points would be helpful to raise the impact of the paper.

Here some suggestions for improvements:

·  The authors use different numbers of animals even in experiments of the same type. Is there a specific reason? Is it derived from statistical analysis (power analysis)? Please, comment.

·        In figure 2, please specify how the VAT secretome is carried out. After how long are the conditioned media collected? This information is relevant to explain the results of subsequent co-culture experiments.

·        Figure 2f-g and 3b do not fully reflect the quantification in figure 2h-i-j and fig 3b. An apparent discrepancy could be due to a misunderstanding in the interpretation of the mice groups in the representative blots. In this case, it needs to be better explained what the numbers and letters above the bands indicate and how to identify the 2 groups of animals used for the study.

·        Similar to the previous point, for figure 5 it is also necessary to specify what the numbers and letters above the bands indicate and what the difference is between 'a' and 'b'.

·        In figure 3e, representative blots are missing. In general, the resolution of this figure should be improved, if possible.

·        Small typing errors are present throughout the text.

Author Response

Sitzia et al. study cardiac remodeling in obesity. Specifically, using both in vivo and in vitro models, they investigated the role of IL-33/ST2 axis in adipose tissue and cardiac tissue and the possible involvement in detrimental fibrotic remodeling of the heart. In addition, the authors proposed a modulation driven by ghrelin.

The manuscript is well written and easy to follow. This is an important study in an area of intense interest; however, elucidation of some points would be helpful to raise the impact of the paper.

Here some suggestions for improvements:

The authors use different numbers of animals even in experiments of the same type. Is there a specific reason? Is it derived from statistical analysis (power analysis)? Please, comment.

We characterized Ten obese non-diabetic male Zucker rats (OB) (fa/fa-) and 10 lean littermates (L) (Fa/+,) 5 obese Zucker hearts and 5 control lean hearts were analysed in molecular and WB analysis (5 were formalin fixed and paraffin embedded). Since we observed high variability in the expression of some genes /proteins (in particular Epac1 and IL-33), we performed and analysed multiple biopsies of the hearts to diminish variability and to increase statistical significance. However, we still observed high variability and we suggested that it was due do the different localization of biopsies (atrial or ventricular) or the prevalence of cell types (fibroblast, endothelium or cardiomyocytes). This point is now better specified in the Material and Methods Section and discussed in the Result Section.

  • In figure 2, please specify how the VAT secretome is carried out. After how long are the conditioned media collected? This information is relevant to explain the results of subsequent co-culture experiments.

H9C2 cells were plated at time zero in the lower chamber of a transwell plate and freshly extracted VAT tissue was directly plated in the upper chamber. This way the media of H9C2 cells were continuously conditioned by secretome of VAT cells. Different wells were plated to collect medium and cells at different time points.  Medium and cells were collected at Tzero, T6h, T24h and T48h: we analyzed all time points and finally we choose for the analysis the cell collected at 48h. This passage is now explained in the Results and in MM Section  

  • Figure 2f-g and 3b do not fully reflect the quantification in figure 2h-i-j and fig 3b. An apparent discrepancy could be due to a misunderstanding in the interpretation of the mice groups in the representative blots. In this case, it needs to be better explained what the numbers and letters above the bands indicate and how to identify the 2 groups of animals used for the study.

Figure and legend are now better explained as suggested by the reviewer.

  • Similar to the previous point, for figure 5 it is also necessary to specify what the numbers and letters above the bands indicate and what the difference is between 'a' and 'b'.

As mentioned above, letters represent multiple biopsies of the same sample: this point is now clarified in the text and figures had been improved with representative blot for each experiment.

  • In figure 3e, representative blots are missing. In general, the resolution of this figure should be improved, if possible.

We apologize for the mistake: the representative blot is now included in the new figure 3.

  • Small typing errors are present throughout the text.

Text has been corrected and spelling-checked for typos.

Reviewer 2 Report

This is an interesting topic with IL-33 pathway, in which there is accumulating evidence about its effect on epithelial/endothelial damage-related phenotypes. The role of Epicardiac fat tissue in obesity is also critical knowledge.  However, this is rather an exploratory investigation, claiming mechanistic data. First of all,  it is. extremely hard to follow the paper's flow and conclusions.  The authors cites literature back and forth for different pathways related to obesity and show some possibly related data. However, it is unclear the role of IL-33/ST2 pathway on cardiac remodeling in obesity. They show that. obese rats have a pro-inflammatory, pro-fibrotic gene expression profile, which is expected and not novel but fails to connect IL-33 pathways into it. First of all, the source of IL-33 is not clear. They need tissue-specific genetic deletions to connect cardiac IL-33 to remodeling and distinguish its detrimental effects on Th2 inflammation on endothelium and lungs.  Also, they don't factor in other adipose-derived molecules. Additionally, it would have been helpful to use inhibitors and activators of cardiac IL-33/ST2 pathways and show both gain of function and loss of function profile and then show the rescue of those phenotypes. Otherwise, they need to rewrite the manuscript with exploratory claims and with better flow. 

Additionally, even their first sentence on obesity prevalence is inaccurate. They should check the most updated data from the CDC website. The obesity rate in the American population is about 42-43%.  

Author Response

This is an interesting topic with IL-33 pathway, in which there is accumulating evidence about its effect on epithelial/endothelial damage-related phenotypes. The role of Epicardiac fat tissue in obesity is also critical knowledge.  However, this is rather an exploratory investigation, claiming mechanistic data. First of all,  it is. extremely hard to follow the paper's flow and conclusions.  The authors cites literature back and forth for different pathways related to obesity and show some possibly related data. However, it is unclear the role of IL-33/ST2 pathway on cardiac remodeling in obesity. They show that. obese rats have a pro-inflammatory, pro-fibrotic gene expression profile, which is expected and not novel but fails to connect IL-33 pathways into it.

First of all, the source of IL-33 is not clear.

As we discussed, IL-33 can be produced by different type of cells and – among them – endothelial cells, adipose tissue and cardiomyocytes. However, we showed an increase of IL-33 in both adipose tissue and in the heart of Zucker rats. Furthermore - when we stimulated the H9C2 cells with adipose tissue - we obtained an increase of IL-33. Indeed, we observed that IL-33 increased with weight in lean rats. In this regard, we hypothesized that adipose tissue releases IL-33 and provokes further release of IL-33 itself by cardiomyocytes. To further support our hypothesis, we measured IL-33 secreted by adipose tissue in co-culture experiment. ELISA performed on cell medium at 6 hours demonstrated a higher amount of IL-33 released by adipose tissue from Zucker rats than those from lean rats (P=0,0019) This experiment is now included in the new Supplementary Figure 1.

They need tissue-specific genetic deletions to connect cardiac IL-33 to remodeling and distinguish its detrimental effects on Th2 inflammation on endothelium and lungs.

Several papers described the protective role of IL33/sST2 pathway in atherosclerosis, type 2 diabetes, obesity, and myocardial remodelling, possibly operating through inhibition of atherosclerotic plaques development. Indeed, in animal model of obesity, it was showed that IL-33 lowered adiposity, decreased fasting glucose concentration, and improved insulin and glucose tolerance.

It is well accepted that IL-33 administration is effective in activating a type 2 cytokine milieu in the damaged heart, consistent with reduced early inflammatory and pro-fibrotic response. However, IL-33 administration in myocardial infarction is associated with worsened cardiac function and adverse cardiac remodelling in animal model, modulating increased infarct size, left-ventricular hypertrophy and cardiomyocyte death (Ghali R, 2020). Intriguingly, to understand the effects of IL-33/sST2 axis on hearts in obesity, it is necessary to better specify the source of the components, to identify the stimuli for their expression and release and to decipher the role of immune cells in this pathway. This way, other studies as those suggested by the referee are necessary to progress diagnostic and therapeutic advances related to IL-33/ST2 signalling.

In this regard - with the in vitro experiments with H9C2 cell - we aimed to highlight the effect of IL-33 signalling on cardiomyocytes to avoid the bias derived by IL-33 effects on immune cells or endothelium. We used different stimuli (such as VAT secretome, ghrelin and leptin) to evaluate the effect of fat accumulation on cardiomyocytes alone. As it concerns the endothelium, histologic analysis did not show increased endothelium expression of IL-33 or morphological change in vessel of myocardium. In last analysis, we also investigated the role of IL-33 in mediating the functions of immune cells. Taking into account the work of Xia (10.1161/CIRCULATIONAHA.120.046789), it was described that Tregs were highly enriched in the myocardium of myocardial infarction mouse model, as they were mainly recruited from the circulation but also accumulated and proliferated according to the expression of the IL-33 and suppression of tumorigenicity 2 genes. Accordingly, we did not observe Treg accumulation while histologic analysis did not reveal area of infiltrated immune cells so we did not further investigate this point.

Finally, our work aimed to evaluate the potential of IL-33/ST2 signalling as a biomarker of heart failure in obesity. So that we focused on cardiac remodelling and we analysed fibrotic pathway both at molecular and protein level. Unfortunately, we did not collect lung tissue of Zucker rats.

Also, they don't factor in other adipose-derived molecules.

Thanks to the co-colture experiments (adipose tissue and H9C2 cells) we evaluated the effect of secretome of adipose tissue on cardiomyocytes focusing on IL-33 signalling. Furthermore, we assessed that this secretome contained IL-33, and we also observed that the amount of IL-33 was higher in adipose tissue secretome from Zucker rats. To evaluate other molecules that could influence this pathway we investigated ghrelin effect on cardiomyocytes, demonstrating an increase of IL-33 expression after cardiomyocytes stimulation. In order to try to answer to the referee’s question, we also determined the effect of leptin stimulation: very preliminary data showed that leptin stimulation reduced IL-33 expression in cardiomyocytes with a opposite trend than those observed after ghrelin stimulation. These data demonstrated that adipose tissue and its secretome are directly involved in IL-33 signalling modulation in cardiomyocytes and that they affect gene expression in cardiomyocyte. These data are now included in Supplementary Figure 2.

IL-33 at T4 hours: CTR Mean 0.6947 ± 0.01324 and Leptin Mean 0.4704 ± 0.06898 with p= p=0,0874.  IL-33 at T48 hours: CTR Mean 1.365 ± 0.1329 and Leptin Mean 1.106 ± 0.1094 with p= 0,1715.

Additionally, it would have been helpful to use inhibitors and activators of cardiac IL-33/ST2 pathways and show both gain of function and loss of function profile and then show the rescue of those phenotypes. Otherwise, they need to rewrite the manuscript with exploratory claims and with better flow. 

We agree with the reviewer that both gain of function and loss of function studies are fundamental to demonstrate the mechanism of action of IL-33/st2 in obesity. However, although several studies existed on IL-33 role in myocardial infarction or cardiac failure and on IL-33 role in glucose tolerance and diabetic complications in obesity (including knock-out models), very little is known about IL-33 cardiac signaling related to obesity. Furthermore, as we discussed in the Manuscript, different results were obtained with different animal models of obesity. In this regard our study is firstly an exploratory one, trying to reproduce human disease in the animal model. We focused on possible inducers of IL-33/ST2 signaling such as ghrelin, leptin and other mediators such as Epac proteins and transcription factor (YY1, MEF2a) and we obtained novel data concerning IL-33/ST2 pathway’s regulation. To confirm our findings, future experiments will be focused on modulating IL-33 signaling with different stimuli and, alternatively, through Epac agonist and antagonist to evaluate their effects. Finally, as we mentioned that genetic background of ZF rats influenced the results we are planning to study IL-33 signaling in a naturally-occurring model of obesity. Thus, we modified the manuscript following reviewer instruction.

Additionally, even their first sentence on obesity prevalence is inaccurate. They should check the most updated data from the CDC website. The obesity rate in the American population is about 42-43%.  

 We apologies for the obsolete citation and we corrected the percentage following updated report from National Health Statistics Reports (NHANES 2021-Number 158, June 14, 2021; dx.doi.org/10.15620/cdc:106273)

Round 2

Reviewer 2 Report

The authors addressed some of my concerns with the same citation methodology rather than experimentation. Still, the story does not flow well and is hard to follow. However, it is much improved from the prior. They should clearly mention about exploratory nature of the study rather than mechanistic claims and think of extra experimentation as future studies.
